# Updates on Larynx Cancer: Risk Factors and Oncogenesis

**DOI:** 10.3390/ijms241612913

**Published:** 2023-08-18

**Authors:** Carlotta Liberale, Davide Soloperto, Alessandro Marchioni, Daniele Monzani, Luca Sacchetto

**Affiliations:** 1Unit of Otorhinolaryngology, Head & Neck Department, University of Verona, Piazzale L.A. Scuro 10, 37134 Verona, Italy; carlotta.liberale@studenti.univr.it (C.L.); daniele.monzani@univr.it (D.M.); luca.sacchetto@univr.it (L.S.); 2Respiratory Disease Unit, University Hospital of Modena, 41124 Modena, Italy; alessandro.marchioni@unimore.it

**Keywords:** larynx cancer, oncogenesis, risk factors

## Abstract

Laryngeal cancer is a very common tumor in the upper aero-digestive tract. Understanding its biological mechanisms has garnered significant interest in recent years. The development of laryngeal squamous cell carcinoma (LSCC) follows a multistep process starting from precursor lesions in the epithelium. Various risk factors have been associated with laryngeal tumors, including smoking, alcohol consumption, opium use, as well as infections with HPV and EBV viruses, among others. Cancer development involves multiple steps, and genetic alterations play a crucial role. Tumor suppressor genes can be inactivated, and proto-oncogenes may become activated through mechanisms like deletions, point mutations, promoter methylation, and gene amplification. Epigenetic modifications, driven by miRNAs, have been proven to contribute to LSCC development. Despite advances in molecular medicine, there are still aspects of laryngeal cancer that remain poorly understood, and the underlying biological mechanisms have not been fully elucidated. In this narrative review, we examined the literature to analyze and summarize the main steps of carcinogenesis and the risk factors associated with laryngeal cancer.

## 1. Introduction

The larynx plays a crucial role in various vital functions, including breathing, swallowing, and speaking. Due to its essential role in producing sound, it is often referred to as the “voice box”. It is composed of a cartilaginous framework, both extrinsic and intrinsic muscles, and a mucosal lining. Laryngeal cancer is the second most common malignancy in the upper aerodigestive tract, following lung cancer [1]. Over 90% of laryngeal tumors originate from the mucosal lining, with the most common cytotype being well-differentiated squamous cell carcinoma [2]. Less frequently, tumors can be chondrosarcomas, leiomyosarcomas, melanomas (2–5% of all laryngeal tumors [3]) or rare neuroendocrine tumors [4]. The clinical behavior of laryngeal cancer varies depending on the site of origin. Supraglottic squamous cell carcinoma (SSC) is commonly associated with cervical lymph node metastasis [5] and often remains asymptomatic until an advanced stage [6]. Subglottic cancer is the rarest type and typically leads to breathing symptoms due to laryngeal obstruction in its narrowest region [7]. Glottic SCC is the most prevalent form of laryngeal cancer, often diagnosed early due to noticeable symptoms such as dysphonia.

The development of laryngeal SCC (LSCC) involves a multistep process starting from epithelial precursor lesions, including dysplasia and intraepithelial lesions [8]. Several factors, such as certain substances or viruses, have been associated with cancer development. Some of these factors have been extensively studied and established as definite risk factors for laryngeal cancer, while others require further research to definitively establish their role as carcinogens, currently being considered as possible risk factors.

According to the latest evidence, epigenetic modifications have been proven to contribute to LSCC development. Particularly, lncRNAs, miRNAs, and mRNAs play a relevant role in cancer development, including differentiation, proliferation, and apoptosis of cancer cells. Moreover, miRNAs may possess tumor-suppressive properties or be oncogenic. Recent studies have demonstrated that miRNAs can also influence resistance to different treatments, such as radiotherapy, chemotherapy, and immunotherapy [9].

## 2. Discussion

### 2.1. Risk Factors

There are numerous risk factors related to laryngeal tumors, particularly squamous cell carcinoma. These factors have been shown to play a significant role in the development of carcinogenesis. Carcinogenesis is a complex and multistep process characterized by progressive cumulative genetic changes and architectural and cytologic alterations of the squamous epithelium [10]. Various agents are involved in this process, leading to mutations and the progression of cellular transformation. Some agents have established cause-and-effect correlations with LSCC, while further research is required for others, and some remain as hypotheses. The following is a brief overview of the main risk factors for LSCC.

#### 2.1.1. Smoking

Smoking tobacco is the most significant risk factor for larynx cancer [11]. Extensive studies conducted since the 1950s have demonstrated the association between tobacco smoke and the development of head and neck cancer [12]. In the United States approximately 95% of laryngeal cancer patients are smokers [13]. The carcinogenic effect of smoking on laryngeal cancer is now well established [14]; however, the specific risk for different types of smoking is less clear. Cigar and pipe smokers, who typically do not inhale smoke, have a less established relationship with the development of laryngeal cancer [15]. Nevertheless, a dose–response relationship has been demonstrated between tobacco smoking and the risk of laryngeal cancer. When alcohol abuse is combined with smoking, the individual risk of developing laryngeal cancer is significantly increased, up to 177 times compared to individuals who neither consume alcohol nor smoke [16]. Therefore, the combined effect of tobacco and alcohol has a multiplicative impact on the risk of developing laryngeal cancer [17].

#### 2.1.2. Alcohol

Alcohol is another significant risk factor for laryngeal cancer, and numerous studies have established the association between alcohol consumption and the development of laryngeal cancer [18]. According to a meta-analysis conducted by the World Cancer Research Fund (WCRF), there is an increased relative risk (RR) of laryngeal cancer with alcohol intake, with a value of 1.09 (95% CI 1.05–1.13) per 10 g of alcohol consumed per day [19]. In a study of Bagnardi et al., significant increases in risk were observed only in moderate and heavy drinkers, with RRs of 1.44 (95% CI 1.25–1.66) and 2.65 (95% CI 2.19–3.19), respectively [20]. This indicates a dose-dependent effect of alcohol on the risk of laryngeal cancer. A recent systematic review by Levesque et al. [21] demonstrated that alcohol intake increases the risk of infectious diseases and cancers, including LSCC, in a dose-dependent manner.

Furthermore, some studies have speculated about a possible interaction between alcohol intake and genetic susceptibility, particularly in relation to genetic polymorphisms of alcohol dehydrogenase (ADH) and aldehyde dehydrogenase (ALDH) enzymes [22].

#### 2.1.3. Opium

Opium is an illicit substance derived from the poppy plant that contains various alkaloids. The International Agency for Research on Cancer (IARC) has classified opium as carcinogenic for humans when it is smoked or ingested in different forms such as raw opium, opium dross, or opium sap. Recent studies have suggested that opium use may be a risk factor for laryngeal cancer, although there is still limited available data on this topic [23]. Preliminary data indicate that regular opium users have an increased overall risk of HNSCC, including laryngeal cancer [24]. However, further research is needed to establish a clearer understanding of the association between opium use and laryngeal cancer.

#### 2.1.4. HPV

Currently, there is a high incidence of high-risk human papillomavirus (HPV) infection in both benign and malignant laryngeal lesions. While the role of HPV in oropharyngeal and genital dysplasia and cancer has been well established by numerous studies, the impact of HPV in laryngeal cancer remains quite unclear [25]. Furthermore, certain studies have highlighted the synergistic role between HPV and other infections such as HIV in the development of oropharyngeal cancer [26]. While smoking remains the primary risk factor for laryngeal cancer, HPV infection is commonly found in nonsmokers and younger patients with laryngeal cancer [27]. Various HPV subtypes have been identified, with some showing stronger correlations with cellular malignant transformation [28]. Among these subtypes, HPV-16 has been frequently observed in laryngeal cancer specimens, as reported in a meta-analysis by Li et al. [27]. HPV-16 has shown a strong association with the development of cancer, including LSCC, and has a relevant role in the clinical prognosis of LSCC tumors [29]. Additionally, HPV-18, another high-risk genotype, is frequently isolated in LSCC [30]. However, the relationship between HPV infection and the risk of laryngeal cancer does not appear to be influenced by factors that affect HPV prevalence. Nevertheless, further research is needed to establish a clear and definitive association between HPV infection and laryngeal cancer risk, considering various influencing factors.

#### 2.1.5. EBV

Brichácek et al. [31] were the first to provide evidence of the presence of the Epstein-Barr virus (EBV) genome and the latent protein EBNA (Epstein-Barr virus nuclear antigen) in malignant cells of laryngeal carcinomas. Subsequently, de Lima et al. [32] conducted a systematic review with meta-analysis to explore the association between EBV and carcinomas of the larynx. The most recent findings suggest a potential role for EBV as a risk or cofactor in the development and/or progression of laryngeal carcinoma.

The role of EBV in LSCC is controversial, as difficulties have been reported in detecting its presence in this type of cancer or it has shown a low prevalence [33]. However, both EBV and HPV are viruses capable of producing oncoproteins that can promote carcinogenesis and tumor progression [34].

Further research, including in vitro studies, is needed to definitively establish and understand this association.

#### 2.1.6. Agent Orange

Among the established risk factors for the development of laryngeal precancerous and cancerous lesions, certain environmental pollutants have been identified. One such pollutant is Agent Orange, an herbicide widely used during the Vietnam War from 1961 to 1971. The correlation between Agent Orange exposure and laryngeal carcinoma has been the subject of recent investigations [35]. Agent Orange contains toxin 2,3,7,8-tetra-chlorodibenzo-p-dioxin (TCDD), commonly known as dioxin, which is a well-known carcinogen [36]. Recent evidence has revealed a relative risk of 1.11 for laryngeal cancer in the exposed cohort of Vietnam veterans, providing a rare high-powered study on this specific topic [35].

#### 2.1.7. *Helicobacter pylori* and Gastroesophageal Reflux Disease

*Helicobacter pylori* (*H. pylori*) is known to play a significant role in the development of various gastrointestinal conditions such as duodenal and gastric ulcers, chronic gastritis, gastric lymphoma, and adenocarcinoma [37]. While its association with these conditions is well established, the relationship between *H. pylori* and laryngeal malignancy has yielded conflicting outcomes in different studies [38]. Recent studies proved that there seems to be a correlation between *H. pylori* infection and laryngeal carcinoma [39]. However, further research with larger patient populations and more comprehensive data is required to confirm this association conclusively.

Regarding the correlation between gastroesophageal reflux disease (GERD) and LSCC, the present evidence remains conflicting. A study by Lewin et al. in 2003 was among the first to examine the relationship between laryngeal dysplasia and early cancer in relation to GERD [40]. Other authors investigated this complex relationship [41,42], but as for *H. pylori*, further studies with larger data are required to prove this cause–effect correlation.

#### 2.1.8. Microbiome

Indeed, the role of the microbiome in laryngeal squamous cell carcinoma (LSCC) has gained attention in recent studies. Gong et al. [43] conducted research that revealed differences in the bacterial composition of the throat between laryngeal cancer patients and healthy individuals. They identified fifteen genera of bacteria, including Fusobacterium, Prevotella, and Streptococcus, which were potentially associated with laryngeal carcinoma. Wang et al. [44] discussed the role of the oral microbiome in oral and oropharyngeal cancer. In a recent study by Yu et al. [45], the role of the oral microbiome in the development of laryngeal cancer has been thoroughly investigated. Fusobacterium was found in several samples of patients with LSCC, similar to its presence in patients with oral SCC [46]. Fusobacterium is an invasive anaerobe, which may lead to chronic inflammation and thus contribute to carcinogenesis. Yu et al. found that shifts in the oral microbiome can act as an early marker of laryngeal cancer, preceding clinical or CT changes [45].

The microbiome, consisting of the community of microorganisms inhabiting our bodies, has been found to have a significant impact on various aspects of host physiology. It can influence the immune system [47], regulate metabolism [48], and even play a role in promoting cancer development [49]. It has been proven that alcohol and tobacco consumption can alter oral microbial composition [50], leading to the selection of microbiota capable of a high rate of carcinogen metabolism. This synergistic role with alcohol and tobacco contributes to promoting LSCC.

The transformation of laryngeal normal mucosa in SCC is a multi-step process, and the microbiome seems to play an active role in this process.

#### 2.1.9. Other Risk Factors

Other risk factors less studied and perhaps less implicated in head and neck cancers are genetic susceptibility, areca (betel) nut chewing, marijuana, diets, lack of vitamins, certain occupations, air pollution exposures, and previous exposure to radiotherapy [40,51,52].

Certain genetic factors may contribute to an individual’s susceptibility to developing head and neck cancers, including laryngeal cancer. Genetic studies have identified specific gene variants that may increase the risk, but further research is needed to fully understand the extent of their influence. The habit of chewing areca nuts is prevalent in certain regions of the world and has been associated with an increased risk of oral and pharyngeal cancers. However, its specific association with laryngeal cancer requires more investigation, being still not clear. The relationship between marijuana use and head and neck cancers, including laryngeal cancer, is not yet well-established. Dietary factors, including a diet low in fruits and vegetables, have been implicated as potential risk factors for various cancers, including head and neck cancers. However, the specific impact on laryngeal cancer requires further investigation. Individuals who have undergone radiation therapy for previous head and neck cancers may have an increased risk of developing a second primary tumor, including laryngeal cancer. However, the risk is influenced by various factors such as the radiation dose, duration since treatment, and individual susceptibility.

Various factors have been identified as potential risk factors, and further studies are needed to establish their role in carcinogenesis and to clarify their real mechanism of action.

### 2.2. Oncogenesis

It is widely assumed that SCC arises from a premalignant progenitor, followed by the outgrowth of clonal cells with cumulative genetic alterations and phenotypic changes that lead to invasive malignancy [53]. For all cancer variants, frequent genetic alterations occur, such as inactivation of tumor suppressor genes and activation of proto-oncogenes through deletions, point mutations, promoter methylation, and gene amplification [54]. These are the main mechanisms of carcinogenesis. Califano et al. [10] delineated a genetic progression model for HNSCC based on the analyses of most frequent genetic alterations in head and neck tumors, shown in Figure 1.

In the early stages of carcinogenesis, there is a loss of heterozygosity (LOH) of 9p21 [55], which is the chromosomal locus where CDKN2A is located. CDKN2A encodes two different transcripts, p16 and p14ARF. The incidence of p16/HPV positivity in LSCC is generally low and varies geographically [56]. However, p16 expression is more commonly observed in non-smokers [54], females [57], and younger patients [58] with LSCC. Both p16 and p14ARF are involved in regulating the G1 cell cycle and mediating the degradation of p53 by MDM2. In LSCC, expression level of p16INK4a is significantly reduced, and hypermethylation has been shown to be a common mechanism causing this downregulation [59]. The loss of function of the p53 tumor suppressor gene plays a crucial role in cancer development [60]. Normally, p53 leads to cell cycle arrest, cellular senescence, and DNA repair when cells are exposed to noxious agents. If DNA repair is ineffective, p53 induces apoptosis to eliminate damaged cells. The inactivation of p16 and p53 can disrupt the normal cell cycle control and promote uncontrolled cell growth. The timing of p53 inactivation in LSCC tumorigenesis is controversial. Some studies demonstrate aberrant expression in early dysplasia, but the majority suggest mutations only late in tumor evolution [61].

Other genes which play a fundamental role in regulating cell cycle are 9p34 (NOTCH1 tumor suppressor or oncogene) [62], 11p15 (HRAS oncogene) [63], 3q26 (PIK3CA oncogene), and 10q23 (PTEN tumor suppressor gene). RAS mutations activate the Raf/MEK/ERK pathway and the PI3K pathway, both of which are involved with cell proliferation, differentiation, and cell survival. RAS mutations and PIK3CA amplifications have been associated with poorer progression-free survival [64].

NOTCH1 has been proven to have both oncogenic and tumor suppressor activities [65]. In LSCC, NOTCH1 acts as a tumor suppressor gene, and most of the mutations are loss-of-function mutations of the epidermal growth factor (EGF)-like ligand or the NOTCH intracellular domain (NICD).

Epidermal growth factor receptor (EGFR) is a member of the tyrosine kinase family of transmembrane receptors. Upon binding with its natural ligands in the extracellular environment, EGFR initiates an intracellular signaling cascade, activating downstream effectors that promote cell proliferation [66]. In some cases, certain cells gain the ability to overproduce EGFR ligands or increase the number of EGFRs on their surface. This creates an autocrine growth pathway, leading to uncontrolled cell proliferation, angiogenesis, and enhanced cell survival [67], as happens in LSCC. EGFR expression usually increases progressively with increasing degree of dysplasia, and it is very elevated in many fully transformed head and neck SCC [68]. High EGFR levels in HNSCC are associated with shorter overall and progression-free survival [69].

Another common genetic alteration observed in the early stages of LSCC carcinogenesis is the loss of chromosome region 3p [70]. These changes in chromosome 3p are associated with the loss of tumor suppressor genes.

Both the dysregulation of EGFR signaling and the loss of the chromosome region 3p are genetic events that promote LSCC.

Late mutations in LSCC include loss of heterozygosis of 17p and point mutations in p53 [71]. These mutations are found in approximately 50% of cells when epithelial dysplasia progresses to invasive carcinoma [72].

Other genes of particular interest in the process of larynx carcinogenesis are 11q13 and cyclin D1. In fact, amplification of 11q13 and overexpression of cyclin D1 are present in more than 45% of head and neck SCC cases and are associated with an increased rate of lymph node metastases and overall poor prognosis [73,74].

Ha et al. [75] demonstrated that the majority of transcriptional alterations in head and neck carcinogenesis occur during the transition from normal mucosa to premalignant lesions. Indeed, according to current evidence, the mutations that occur in the transition from carcinoma in situ to invasive carcinoma are far fewer than those recorded in the early stages of carcinogenesis.

One of the latest mutations concerns cell adhesion proteins. E-cadherin (CDH1) is a surface protein mediating cell–cell contact. Loss of cell–cell contact has been associated with tumor invasiveness and metastatic potential in many cancers [76]. The mechanism of E-cadherin inactivation in LSCC is still not known.

### 2.3. The Emerging Role of lncRNAs, miRNAs, and mRNAs

miRNAs are a type of non-coding RNA (ncRNA) that plays a crucial role in a wide variety of biological processes [77]. Based on the latest research, it has been shown that alterations to miRNAs are what initially set the stage for the development of cancer [9]. Molecular changes found in carcinogenesis, such as transcriptional deregulations, chromosomal editing, protein abnormalities, and so on, can cause modifications in miRNAs. In a recent study, Hegazy et al. [9] stated that the advancement of LSCC is influenced by the alteration of miRNA expression. In particular, laryngeal cancer cells show downregulation of miR-138, which leads to less apoptosis and more cell proliferation [78]. Other miRNAs that play a relevant role in the development of laryngeal cancer (LC) are miR-195-5p and miR-206 [79]. The first one has a role in proliferation, migration, invasion, and epithelial–mesenchymal transition of laryngeal cancer cells, while the second one has been proven to be associated with poor prognosis and distant metastasis. Additionally, an elevated level of miR-23a correlates with lymph node metastasis and poor prognosis [80]. miR-106a-5p accelerates the development of LC [81], as does the reduction in miR-107 [82]. These are just a few examples of the miRNAs known to be involved in laryngeal carcinogenesis to date. In fact, there are many other miRNAs that play relevant roles in the laryngeal carcinogenesis process, which are currently being studied and investigated. The research field of miRNAs is developing rapidly and gaining more and more interest due to their potential role as biomarkers and possible therapeutic targets. As demonstrated by Hegazy et al., specific miRNAs are widely recognized to be associated with tumor occurrence, even in their earliest stages, or with a worse prognosis. Consequently, miRNAs could serve as promising candidates for biomarkers [9].

Recent studies have highlighted the significance of long non-coding RNAs (lncRNAs) as crucial regulators of miRNA expression in LSCC. Conversely, miRNAs can also target and regulate the expression of lncRNAs. Furthermore, miRNAs can exert their effects on downstream messenger RNAs (mRNAs) through transcriptional or post-transcriptional mechanisms, thereby influencing various physiopathological processes in LSCC [83]. The intricate interplay between lncRNAs, miRNAs, and mRNAs forms complex cross-regulatory networks that play a significant role in the tumorigenesis and progression of LSCC. These networks hold promise as potential biomarkers and therapeutic targets in LSCC research. Tumor cells exhibit distinct cell cycle patterns to sustain their proliferation, differentiation, and other vital physiological processes. Various lncRNAs, miRNAs, and mRNAs have been identified as active participants in the regulation of cell cycle progression, influencing the development of different types of tumors, such as LSCC [84].

Apart from their well-established role in cancer development and potential as biomarkers, lncRNAs, miRNAs, and mRNAs have also been implicated in the development of resistance to LSCC therapies. Consequently, a growing number of studies have been directed towards exploring lncRNA–miRNA–mRNA networks and unraveling the molecular mechanisms underlying chemoresistance [85]. Furthermore, other studies have highlighted the potential of lncRNAs, miRNAs, and mRNAs in the development of radiotherapy resistance [86].

It is essential to identify prognostic biomarkers with high sensitivity and accuracy to predict clinical outcomes and guide timely treatment for patients with LSCC. Ongoing research in the genetic field has revealed that the expression levels of lncRNAs, miRNAs, and downstream target mRNAs are associated with the prognosis of LSCC patients, as demonstrated by an increasing number of experimental results [84].

The intricate interplay between lncRNAs, miRNAs, and mRNAs holds promising insights improved strategies for the diagnosis, treatment, and prognosis of LSCC.

### 2.4. Clinical Correlations of Oncogenesis

Following the process explained in Figure 1, LSCC has numerous histologic precursors, including leukoplakia, erythroplakia, squamous hyperplasia, and squamous dysplasia.

Leucoplakia is a descriptive term for mucosal white plaques. It does not denote a specific pathologic entity. This term is used to described clinically a mucosal lesion of the larynx. Both benign and dysplastic vocal fold leukoplakia lesions carry some risk of malignant transformation [87]. Leukoplakia is the most common potentially malignant disorder of head and neck cancer. The range of malignant transformation incidence from leukoplakia to squamous cell carcinoma amounts from 0% to 64.7% for laryngeal leukoplakia [88]. This wide range of incidence depends on several factors, such as diagnosis criteria, geographical locations, different populations, and other factors [89]. The potential for malignancy is usually determined based on the histology, especially the grade of dysplasia. Several studies demonstrated that moderate or severe dysplasia leads to a much higher risk of cancer evolution than mild dysplasia [90]. To date, unfortunately, clinical detection is not as accurate as histology in predicting the risk of malignancy. The current management of laryngeal leukoplakia involves both a “watch and wait” approach and several biopsies, especially for patients with an oncologic personal history or relevant risk factors [91]. In fact, only the histologic results of biopsies can confirm the presence of dysplasia and guide the therapeutic management of patients.

Laryngeal erythroplakia and leukoerythroplakia (leukoarythroplasia) represent red plaque lesions and white-with-red speckled plaque lesions secondary to increased hypervascularity and vascular dilation. Erythroplakia is considered the most important clinical predictor for either presence of malignancy or progression toward carcinoma [92]. It is also considered to carry dysplastic and malignant potential [93]. Clinically, erythroplakia appears as a red spot due to the increased blood vessels under the epithelial surface. Erythroplakia and aberrant microvasculature are similarly accentuated using narrow band imaging, which is a very useful tool in clinical practice to detect suspicious laryngeal lesions.

Hyperplasia and dysplasia, on the contrary, are pathologic entities. Squamous hyperplasia is a reactive phenomenon consisting of a thickened squamous epithelium. Hyperplasia includes both acanthosis (increased thickness of the spinous layers) and expansion of the basal cell layer (increase in basal/parabasal cells). When this phenomenon is localized, it appears clinically as leukoplakia.

Squamous dysplasia is, instead, a premalignant proliferation of squamous epithelium. It is recognized as the precursor of SCC. The diagnosis and grading of dysplasia are based on the assessment of a constellation of features related to both nuclear atypia as well as architecture and maturation of the cells. The World Health Organization (WHO) Classification of Head and Neck Tumors endorses a two-tier system for the grading of laryngeal dysplasia [94]. The two grades of dysplasia are either low-grade dysplasia/squamous intraepithelial lesions or high-grade dysplasia/squamous intraepithelial lesions based on the degree of cytologic and architectural aberration.

Considering the current understanding of pre-neoplastic lesions and their potential progression into cancer, early intervention is vital. Early diagnosis plays a fundamental role in managing neoplastic conditions, including LSCC. Detecting laryngeal cancer at an early stage allows for organ-preserving approaches, leading to better functional and oncological outcomes for patients. Integrating clinical and histological diagnosis is crucial, and prompt intervention is essential when malignant histological changes are identified.

## 3. Conclusions

Laryngeal cancer is one of the most frequent tumors of the upper aero-digestive tract. While certain factors such as alcohol consumption and tobacco smoking have been established as risk factors for laryngeal cancer, the correlation of many other risk factors with the mechanism of carcinogenesis remains unclear. The process of laryngeal carcinogenesis has been well described; however, there are still aspects that lack understanding, and the underlying biological mechanisms have not been fully elucidated. Indeed, the emerging role of lncRNAs, miRNAs, and mRNAs is becoming increasingly important in our understanding of cancer development and its clinical and therapeutic implications. These non-coding RNAs have been shown to play critical roles in various aspects of cancer biology, including tumor initiation, progression, metastasis, and response to treatment. Gaining insight into the molecular mechanisms involved in carcinogenesis could enable the prediction of tumor progression and facilitate targeted interventions at various stages of the disease. The ultimate goal is to develop precise and effective therapies through a precision medicine approach. Achieving this objective requires further comprehensive studies on the molecular biology of cancer and the role of different risk factors.

## Figures and Tables

**Figure 1 ijms-24-12913-f001:**
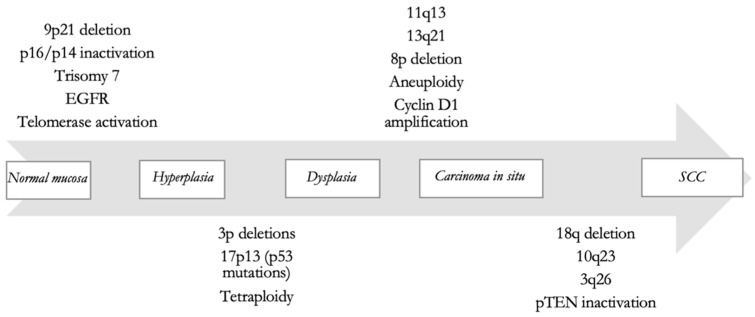
Hypothetical model for HNSCC carcinogenesis, from Califano et al. [10].

## Data Availability

Not applicable.

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
