# Peer review of "Updates on Larynx Cancer: Risk Factors and Oncogenesis"

_ijms, 2023, doi:10.3390/ijms241612913_

Round 1
Reviewer 1 Report
Liberale at al. submitted the manuscript entitled: "Updates on larynx cancer: risk factors and oncogenesis".
The manuscript is well written, clear and easy to understand. The references selected for each section are appropriate. I personally like the concept of writing Introduction as firs section and going immediately to Discussion section where various aspects are covered. No need for too many subsections.
Only comment would be in the line 145/146 to write strains in Italic form.
Other than that, the manuscript represents a very nice and informative output for the researchers as well as for medical doctors.
Author Response
Dear Editor,
Dear Reviewers,
We sincerely appreciate your comments and suggestions aimed at enhancing our work. We are delighted that you found merit in our efforts and recognized the strengths of our narrative review on a profoundly intricate subject. We have diligently incorporated all the recommended changes.
Reviewer 1: we have rectified the lines 145/146 as suggested.

Reviewer 2 Report
Sometimes it is nice to read a review because it gives a nice overview without having to do it yourselves. The paper «Updates on larynx cancer: risk factors and oncogenesis» fulfills most of those expectations, but there are some issues.
The abstract is not very well structured, and need to be “tightened up”. The line 16-18 should perhaps be at the end, and maybe no need for line 21-23? There is a spelling mistake in line 17.
From the introduction the authors go straight to the discussion. I would have inserted a small paragraph about the method.
The discussion is fine. Some of the references are older, but as some of the topics are not very “hot” there are not many newer references. Some topics are not so relevant for LSCC, but find it nice that they have been discusses (like HP). Not sure that too much effort should be placed in further and larger studies though.
The oncogenesis is complex, and is for those with special interest, and so is the case also for Inc-RNAs, miRNAs and mRNAs. Nice with a clinical correlation to oncogenesis. NBI = narrow band imagine? Do not use only the abbreviation. Spelling mistake in line 355.
ok
Author Response
Dear Editor,
Dear Reviewers,
We sincerely appreciate your comments and suggestions aimed at enhancing our work. We are delighted that you found merit in our efforts and recognized the strengths of our narrative review on a profoundly intricate subject. We have diligently incorporated all the recommended changes.
Reviewer 2: we have revised the abstract and addressed spelling errors. As mentioned, certain aspects covered may not be as current, leading to limited recent references beyond our own.
We have not included a concise paragraph concerning the methodology. Our review is narrative in nature, we acknowledge that a systematic approach like PRISMA was not employed, requiring a specific section. Therefore, we maintained the original structure of the article, without introducing a dedicated methods section.

Reviewer 3 Report
The topic of the article is very interesting. When dealing with infection issues, I suggest that the authors consider and cite the prevalence of HPV infection in the pharynx and the risk factors for HPV persistence (for example bacterial co-infections) as described by other Italian studies. Moreover, metabolic risk factors should be considered:
Hidalgo-Tenorio C, Calle-Gómez I, Moya-Megías R, Rodríguez-Granges J, Omar M, López Hidalgo J, García-Martínez C. HPV Infection of the Oropharyngeal, Genital and Anal Mucosa and Associated Dysplasia in People Living with HIV. Viruses. 2023 May 15;15(5):1170. doi: 10.3390/v15051170. PMID: 37243256; PMCID: PMC10222174.
Ciccarese G, Herzum A, Rebora A, Drago F. Prevalence of genital, oral, and anal HPV infection among STI patients in Italy. J Med Virol. 2017 Jun;89(6):1121-1124. doi: 10.1002/jmv.24746. Epub 2016 Dec 23. PMID: 27935070.
Drago F, Herzum A, Ciccarese G, Dezzana M, Pastorino A, Casazza S, Nozza P, Rebora A, Parodi A. Prevalence and persistence of oral HPV infection in Italy. J Eur Acad Dermatol Venereol. 2019 Apr;33(4):e150-e151. doi: 10.1111/jdv.15380. Epub 2019 Jan 1. PMID: 30520177.
Huang J, Chan SC, Ko S, Lok V, Zhang L, Lin X, Lucero-Prisno DE 3rd, Xu W, Zheng ZJ, Elcarte E, Withers M, Wong MCS; NCD Global Health Research Group, Association of Pacific Rim Universities (APRU). Disease burden, risk factors, and trends of lip, oral cavity, pharyngeal cancers: A global analysis. Cancer Med. 2023 Jul 30. doi: 10.1002/cam4.6391.
References in the reference list have double numbers.
Author Response
Dear Editor,
Dear Reviewers,
We sincerely appreciate your comments and suggestions aimed at enhancing our work. We are delighted that you found merit in our efforts and recognized the strengths of our narrative review on a profoundly intricate subject. We have diligently incorporated all the recommended changes.
Reviewer 3: we have delved deeper into the topic of HPV infection by incorporating additional data, along with two of the recently suggested references. Furthermore, we have rectified the reference list, which had duplicated reference numbers.
We extend our heartfelt gratitude for dedicating your valuable time to review our study. We trust that our enhancements align with your expectations.
